# Disaster Evacuation for Home-Based Patients with Special Healthcare Needs: A Cross-Sectional Study

**DOI:** 10.3390/ijerph192215356

**Published:** 2022-11-21

**Authors:** Yukari Matsumoto, Hisao Nakai, Yumi Koga, Tamayo Hasegawa, Yumiko Miyagi

**Affiliations:** 1School of Nursing, Faculty of Medicine, Fukuoka University, Fukuoka 814-0133, Japan; 2School of Nursing, Kanazawa Medical University, Kahoku 920-0265, Japan

**Keywords:** disaster preparedness, emergency preparedness, evacuation intention, vulnerable populations, pediatric patients receiving home medical care

## Abstract

Recent super-typhoons and torrential rains triggered by global warming have had disproportionately large effects on medically vulnerable people in Japan. This study aimed to identify factors associated with intention to evacuate to the nearest public shelter among family caregivers of pediatric patients receiving home medical care. The study included family caregivers of these patients from the Department of Pediatrics, Fukuoka University Hospital, Japan, including family caregivers of young adults with special healthcare needs. An original questionnaire was prepared drawing on previous studies and used for an interview survey. Overall, 57 individuals provided valid data and were included in the analysis. Factors associated with evacuation intention were non-use of a home ventilator (odds ratio [OR] 3.99, 95% confidence interval [CI]: 1.13–14.03) and not having made arrangements to use a non-public shelter (OR 7.29 95% CI: 1.62–32.88). This means that those who use mechanical ventilation or have secured alternative places to go if they need to evacuate their homes may not use the nearest public shelter in a disaster. We recommend that policy makers consider the use of mechanical ventilation and the availability of non-public shelters as predictors of evacuation behavior when considering disaster preparedness for these patients.

## 1. Introduction

Local torrential rain events caused by tropical cyclones fueled by global warming have occurred increasingly often in recent years [1]. Heavy rains from zonal precipitation systems frequently occur in Japan from early summer to autumn, resulting in serious disasters such as landslides, flooding, and mudslides, and causing severe damage [2,3]. This type of torrential rain is reportedly caused by a linear-stationary precipitation system called “Senjo-kousuitai’’ in Japanese [4,5]. These precipitation systems are frequently concentrated in Kyushu, the Nansei Islands, Kii Peninsula, and on the Pacific sides of Shikoku, with fewer appearances in northern Japan and the Sea of Japan side of the Japanese Islands [4]. Since 2000, damage caused by heavy rain occurring as a result of “Senjo-kousuitai” has been observed in northern Kyushu. During the torrential rains in the region in July 2017, more than 40 people were killed [4] and approximately 6000 homes, mainly in Asakura City, lost power for at least 3 days [6]. Prolonged power outages due to natural disasters have a serious impact on older adults and medically vulnerable people [7]. Climate change has also had disproportionate effects on the health of children in recent years [8,9]. These disasters may be life-threatening, especially for children dependent on technology such as mechanical ventilation or suction devices [10]. Prolonged power outages following disasters threaten the lives of these children. The Tohoku region experienced the most extensive blackout after the Great East Japan Earthquake in 2011. The average power outage experienced by children dependent on technology in this instance was 7.8 days, and 55% of them were evacuated to hospital because of the effects of the power outage [11]. Similarly, Gillen et al. found that approximately half of the parents of these children who experienced a power outage called or visited the hospital with the child when the power outage occurred.

Global damage from recent torrential rains and super-typhoons poses a major threat not only to children who are dependent on medical technology but also to medically vulnerable adults, including older adults. Past disasters have shown that treatment for existing or new symptoms that occur after a disaster in medically vulnerable populations is often delayed in the post-disaster period [12]. It is therefore important to ensure that evacuation plans and destinations are prepared, especially for medically vulnerable people.

Previous studies on evacuation intentions found that the willingness to evacuate in the event of a hurricane evacuation order decreases with age [13,14]. Women are also more likely than men to evacuate because they are more aware of the roles of long-term care and incentives for evacuation [15]. A survey of children and families affected by Hurricane Irma in 2017 found that mothers who experienced evacuation, chaos and loss due to Irma were more willing to evacuate again than those who did not [16]. After Hurricane Matthew, it was reported that those with a close-knit community they could rely on during the disaster did not evacuate and stayed with friends and relatives [17]. The presence of children in the home is well known to influence parents’ evacuation decisions [18]. Households with children with disabilities tend to stay in place in the event of a disaster, and do not evacuate even if evacuation is possible [19]. Children with disabilities are more difficult to evacuate from disasters, as are those on ventilators and suction devices. Children with mobility impairments may not be able to squat under their desks during an earthquake, hike hillsides during a flash flood, or run to an evacuation area on high ground in the event of a tsunami [19]. Children with cognitive impairments may also not be able to recognize danger signs or understand imminent threats [20]. To optimize the evacuation of vulnerable groups, it is therefore necessary to identify vulnerable people, mobilize social support such as family and friends, and formulate individual evacuation plans [21]. 

The Japanese government, following trends in other countries, recommends that any children receiving medical care at home and all older adults should be routinely enrolled with the emergency evacuation assistance registry [22,23,24]. When there is a risk of a disaster, the government classifies the risk by five levels of vigilance, calling for early evacuation of medically vulnerable people and older adults [25]. 

The COVID-19 pandemic has prompted the Japanese government to call for “decentralized evacuation” to the homes of relatives and acquaintances rather than to public shelters, to avoid the risk of infection by indiscriminate crowding and close contact [26,27]. The pandemic has also affected the decision-making of medically vulnerable individuals about evacuation. Fear of COVID-19 may make medically vulnerable people hesitate to evacuate even in disasters where evacuation is essential [22]. A recent survey of willingness to evacuate during hurricanes found that more than half of respondents might not evacuate when there is a hurricane forecast, and 70% of these were concerned about COVID-19 infection [28]. 

In Japan, the number of pediatric patients with home medical devices is constantly increasing; it was estimated to be 20,000 in 2020 [29]. The importance of building evacuation shelters for vulnerable people has been discussed [30,31], but a system that allows pediatric patients receiving home medical care to evacuate and receive suitable care is still under development. These patients and their families must therefore make their own decisions whether to evacuate to a public shelter in the event of a disaster, and also make their own plans for survival after evacuation. The aim of this study was to determine the factors associated with the intention to evacuate to the nearest public shelter in the event of a disaster among parents with children requiring medical devices or procedures, and visiting the Pediatrics Department of Fukuoka University Hospital, Japan.

## 2. Materials and Methods

### 2.1. Terms Used in Study

#### 2.1.1. Public Shelter

In Japan, local governments designate public shelters, which are facilities where residents can escape the dangers of disasters and live during their evacuation. They publish a list of these public shelters. For example, a list of target areas can be obtained from each municipality’s website [32]. Public shelters include schools and community centers [33], with 92.1% of public schools designated as evacuation centers [34]. However, there may be problems with using public shelters. For example, after the Great East Japan Earthquake of 2011, a large number of victims were forced to live in evacuation shelters for a long period of time, resulting in a decline in mental and physical function and illness [33]. 

#### 2.1.2. Welfare Shelter

In Japan, facilities to which vulnerable people can evacuate are called welfare shelters. Local governments require existing facilities for older adults to set up shelters separate from the public shelters to provide space for older adults, children, and medically vulnerable people who need to evacuate their homes [31,35].

#### 2.1.3. Pediatric Patients Receiving Home Medical Care

We included children aged ≤15 years requiring home medical care and patients with healthcare transition needs [36] visiting pediatric departments, i.e., adults with special healthcare needs [37,38].

### 2.2. Data Collection

The study included family caregivers of pediatric patients receiving home medical care who were receiving care on an outpatient basis from Fukuoka University Hospital. Family caregivers included caregivers of adults with special healthcare needs who continued to receive treatment in pediatric departments. The survey method was an interview survey carried out in conversation with one of two nurses (one researcher and one outpatient nurse). To ensure consistency, we explained the content of the survey to the outpatient nurse, who also received training in listening. Participants were 57 family caregivers of these outpatients drawn from the Pediatrics Department of Fukuoka University Hospital, Fukuoka, Japan. The sample selection used non-probabilistic sampling. Fukuoka University Hospital has the largest perinatal maternal and child medical center in western Japan, accepts most of the relevant patients for this study, especially from western Japan, and provides ongoing treatment and family support for premature babies and children with congenital diseases [39,40]. Before starting the survey, participants were asked to read information about the purpose of the survey and the confidentiality policy and provide written informed consent.

For the survey, we developed a questionnaire-based interview survey that assessed the preparedness of family caregivers of these patients for disasters, the hazards assumed at their place of residence, knowledge required for evacuation, and willingness to evacuate. The questionnaire development was based on studies by Tanaka [41] and Nakagawa et al. [42] on disaster preparation and plans for Japanese children with severe disabilities. The survey was conducted from 26 May 2021 to 22 December 2021.

### 2.3. Survey Content

#### 2.3.1. Background Information about Pediatric Patients Requiring Home Medical Care

The survey asked about the children’s age, number of cohabitants, sex and diagnosis.

#### 2.3.2. Medical Devices and Procedures

The survey was used to collect information about the use of mechanical ventilation, tracheostomy, oxygen inhalation, suctioning, tube feeding, urethral catheterization, blood glucose measurement, nasotracheal airway, central venous nutrition, inhalation, and mechanically assisted coughing.

#### 2.3.3. Family Caregiver Background

We also sought information about the family caregiver, including age, sex, employment status (regular or non-regular employment; Yes/No), and previous experience of evacuation during a disaster (Yes/No).

#### 2.3.4. Preparation in the Event of a Disaster

Items about whether the caregiver was prepared for disaster included whether they had prepared children’s clothing, food, equipment and medication for the patient, and if they had secured a power supply to ensure that they could continue to use medical devices in the event of a power outage. These were all answered “Yes/No”.

#### 2.3.5. Destination of, Preparation for, and Intention to Evacuate

Questions about whether the caregiver had arranged an evacuation site other than general shelters, and intention to evacuate to welfare shelters were answered “Yes/No”. A question about whether they were mentally prepared for evacuation life had answer options of “Yes”, “Somewhat”, “Not really” and “No”, and whether they had requested help for evacuation support and registered with the ledger on support required in the event of a disaster held by the local government were answered “Yes/No”.

#### 2.3.6. Knowledge of Welfare Shelters, Decentralized Evacuation

We asked whether the caregiver knew the location of a designated shelter, about welfare shelters, the location of a welfare shelter, and the proposal for decentralized evacuation, all of which were answered “Yes/No”.

#### 2.3.7. Disaster Risk in Residential Areas

Finally, we asked whether they knew about the risk of river flooding, landslides, and tsunami flooding in the area of their residence.

### 2.4. Analytical Methods

The analysis included 57 people who provided complete and valid answers to all items related to the background of the patient and family caregiver, medical devices and procedures required for the patient, preparedness for disasters, destination, preparation and intention of evacuation, knowledge of welfare shelters, knowledge about decentralized evacuation, and disaster risk in their area of residence.

To understand the characteristics of the participants, we assessed the median age (and range) of the patients. The distribution of age of the family caregivers was also determined. We also examined the distribution of medical devices used by and medical procedures required by patients.

To evaluate the factors affecting intention to evacuate to a designated shelter, patients were divided into two groups by age (under or over 15 years of age) to fit the definition used in pediatric care in Japan [43]; by whether they lived with more or fewer than four cohabitants (using the number of people per household in the 2020 census) [44]; and by whether the family caregiver was under or over 30. We then examined the relationship between each factor and intention to evacuate [13,14]. We used the χ^2^ test or Fisher’s direct probability test to determine the relationship between intention to evacuate to a public shelter and the background of the patient and family caregiver, medical devices and procedures required, preparedness in case of a disaster, intention and plan for evacuation in case of a disaster, request for evacuation support, and mental preparation for evacuation life. To evaluate the impact of the pediatric patients’ use of medical devices, necessary medical procedures, disaster preparedness, request for support in a disaster, and preparedness for evacuation life on intention to evacuate to designated shelters, we used the following methods. Age classification of the patient and experience of disaster by the family caregiver [16,45] were entered by forced entry. Items with significance of *p* < 0.05 in the univariate analysis, i.e., working status of the caregiver, use of ventilators, arranging an evacuation site other than the designated shelter, and number of cohabitants, were analyzed in a stepwise binomial logistic regression analysis. Each variable was entered after checking multiple collinearities (variance inflation factor ≥ 10). The significance level was set at 5%. SPSS Ver27 (IBM Corporation, Armonk, NY, USA) was used for all statistical analyses.

### 2.5. Ethical Considerations

This research was conducted in accordance with the Declaration of Helsinki, 1995 (as revised in Seoul, 2008) and carried out with the consent of the university medical research ethics review committees at the authors’ universities (U21-04-010, I469). Participants were given written and verbal explanations of informed consent. They received an explanation of the purpose and importance of the survey, the survey method, voluntary participation, and the anonymity of participants’ responses, and agreed to sign a consent form.

## 3. Results

Overall, 89 people agreed informally to cooperate with the survey, but the risk of infection due to COVID-19 meant that 32 of them gave up the interview survey because they postponed medical examinations or switched to online consultations. Responses were obtained from 57 people who continued to visit the hospital on an outpatient basis during the COVID-19 pandemic (64.0% response rate). The data of 57 participants (64.0%) were therefore included in the analysis.

The median (range) age of the pediatric patients was 7 (0–46) years, 48 (84.2%) were under 15 years old, and nine (15.8%) were 15 years old or older. In total, 29 (50.9%) were male and 28 (49.1%) were female, and the number of cohabitants was four or more for 41 (71.9%) and fewer than four for 16 (28.1%) participants.

Respondents could choose multiple answers for the medical devices and procedures required for the patients. Overall, 47 patients (78.9%) required tracheal suctioning, 40 (70.2%) required tube feeding, and 39 (68.4%) oxygen inhalation. Figure 1 shows the medical devices and procedures required by these patients.

Overall, 54 family caregivers (94.7%) were in their thirties or older, and three (5.3%) were in their twenties or younger. A total of 56 (98.2%) were women and one (1.8%) was a man; 20 (35.1%) were working, of whom 11 (19.3%) were non-regular employees and nine (15.8%) were regular employees. Five (8.8%) had experienced a disaster in the past (Table 1).

The most common items for disaster preparedness included a power supply for continuing use of medical devices during power outages (available to 35, 61.4%), food for the patients (18, 31.6%), and medical hygiene materials (16, 28.1%). A total of 25 (43.9%) respondents said that they would evacuate to a designated shelter and 47 (82.5%) that they would evacuate to a welfare shelter. Overall, 25 (43.9%) were registered with the support ledger of the local government, 20 (35.1%) had secured evacuation sites other than general evacuation centers, 19 (33.3%) had requested help from others to provide evacuation support, and nine (15.8%) had planned for life after evacuation. Knowledge of welfare shelters and decentralized evacuation was as follows: 51 (89.5%) knew the location of the designated shelter, 21 (36.8%) knew that the government had called for decentralized evacuation, 11 (19.3%) knew about welfare shelters, and two (3.5%) knew the location of a welfare shelter. In terms of risk of disaster in residential areas, 46 (80.7%) were aware of the risk of river flooding, and 42 (73.7%) about the risk of landslides and tsunami flooding (Table 1).

Among the respondents, the 27 people (65.9%) (*p* = 0.018) who lived with at least four cohabitants, the 19 (70.4%) (*p* = 0.04) whose patient used a home respirator, and the 15 (75.0%) (*p* = 0.035) who had arranged an evacuation site other than the general shelter were significantly more likely to respond that they were not intending to evacuate to a designated shelter.

A binomial logistic regression analysis was performed on factors associated with intention to evacuate to a designated shelter. The factors associated with the intention of evacuating children receiving home medical care to a public shelter were non-use of a home ventilator (OR 3.99, 95% CI: 1.13–14.03), and not having arranged an evacuation site other than the designated shelter (OR 7.29 95% CI: 1.62–32.88) (Table 2).

## 4. Discussion

Of the children receiving home medical care who participated in this study, 15.8% were young adults with special healthcare needs. The number of patients in healthcare transitions requiring special medical care is increasing, and the collaboration between pediatrics and other clinical departments is progressing [46]. According to a 2016 survey of five hospitals in Japan, the proportion of young adults with special healthcare needs aged 14 years or older attending the hospital ranged from 11.4% to 21.6%, suggesting that the percentage of young adults with special healthcare needs in this study [47] was typical.

According to a previous study in Japan, the gender distribution of pediatric patients with special healthcare needs was about 52% male and 46% female, with an average age (standard deviation) of 7.8 (5.5) years [48]. According to a survey by Kitakyushu, the median age of fathers and mothers of pediatric patients with special healthcare needs was 40 years old, and 42.7% of mothers worked [49]. The pediatric patients in this study were 50.9% male, the median age was 7 years old, 94.7% of family caregivers were 30 years old or older, and 35.1% were working. It is therefore likely that the participants of this study were reasonably representative of the overall study population. A nationwide survey by the Ministry of Health, Labour and Welfare found that 74.4% of pediatric patients with special healthcare needs used tube feeding, 69.0% suctioning, 41.8% endotracheal intubation/tracheostomy, 40.1% inhalation, 37.5% oxygen inhalation, and 33.0% home ventilators [50]. In this study, tube feeding and suctioning were the most frequent needs of patients, followed by oxygen inhalation. Considering the differences in target age, survey content, and survey methods, it is possible that the distribution of privately-used medical devices and medical procedures followed a similar pattern to the national survey.

Overall, 56.1% of family caregivers of pediatric patients receiving home medical care were not intending to evacuate to a public shelter. Family caregivers of children using ventilators may be less likely to evacuate than those of children who do not use this equipment. The results are consistent with a previous study of power-dependent home care patients in Japan [51]. In the Great Hanshin-Awaji Earthquake of 1995, families with severely disabled children temporarily evacuated to public shelters, but many were forced to leave the public shelters and stay at relatives’ homes, and evacuate the child to hospital. Some families even had to live in their cars because (1) they required a variety of specialized resources and (2) children with intellectual disabilities showed a variety of behavioral problems. Some families were forced to evacuate by car [52]. This situation was similar after the Great East Japan Earthquake of 2011, when children using mechanical ventilation sought refuge in cars instead of public shelters [53]. Even in the 2016 Kumamoto earthquake, those who needed medical attention gave up going to a shelter out of fear of causing inconvenience to others and to avoid difficulties in continuing care because of lack of private space [54]. Ventilator users may therefore have judged that they would not be able to function in a public shelter. However, 82.5% of family caregivers in this study said they would evacuate to a welfare shelter. In Japan, welfare shelters are shelters for old, disabled, and medically vulnerable people in the event of a disaster. They are installed in existing facilities for older adults and people with disabilities [35]. However, there are no doctors or nurses on the premises because these are not medical facilities. They also have existing residents. In the Kumamoto earthquake, welfare shelters were available, but there were no additional staff available to care for the temporary shelter-seekers because of the disaster-related shortage of nursing care workers [55]. The current system of welfare shelters therefore has significant flaws, and there may not be sufficient capacity to provide adequate care for medically vulnerable people who safely evacuate to a welfare shelter. A survey after the Great East Japan Earthquake revealed that emergency plans for disaster preparedness in welfare shelters, including staffing, cooperation with medical professionals [56], and accommodation for the diverse needs of pediatric patients receiving home medical care and older adults were all lacking [53]. However, the high willingness of family caregivers in this study to evacuate to welfare shelters may suggest that they have higher expectations of welfare shelters than public shelters, without fully understanding the characteristics and challenges of welfare shelters.

This study had several limitations. First, the participants were family caregivers of pediatric patients receiving home medical care and care on an outpatient basis at one university hospital in Fukuoka Prefecture, and sample size calculations were not performed. We could not control for the effect of the sex of family caregivers, a potential confounding factor, because this study prioritized collecting data on the children, rather than caregivers. Survey results can also be influenced by their timing, in this case perhaps by whether there had been a recent disaster. It is possible that there were more vulnerable participants among the 32 who were unable to participate in the study because of fears of exposure to COVID-19. No comparison was made from before the COVID-19 pandemic, and we therefore do not know the impact of the pandemic on people’s intention to evacuate. It is also possible that the presence or absence of power dependence affected evacuation intention. Given these limitations, the results of this study cannot be generalized. Family caregivers of relatively new patients, or who only recently began living with the patient, might also have been less likely to respond to the survey because they already felt under pressure. This would mean that family caregivers who were more experienced and used to living with their patient’s needs may have been over-represented in our sample. To increase reliability, future studies should carry out sample size calculations, increase the number of participants, and measure internal consistency using the reliability coefficient. Finally, this study had a cross-sectional design and it is therefore not possible to establish any causal relationships between the variables under investigation.

## 5. Conclusions

University hospital staff and municipal policy makers should be aware that family caregivers of pediatric patients receiving home medical care who use ventilators may not evacuate to public shelters even if they are opened. Estimated evacuation rates of these patients to public shelters may therefore not accurately predict actual evacuation rates. It is recommended that staff check if family caregivers of these patients have secured other evacuation sites beforehand, and that their arranged evacuation sites will be suitable. In particular, many children using medical devices such as mechanical ventilators may have judged that it will be difficult to continue mechanical ventilation in public shelters. Considering the current challenges of Japanese welfare shelters, we recommend considering the installation of a new evacuation shelter dedicated to the evacuation of pediatric patients requiring mechanical ventilation.

## Figures and Tables

**Figure 1 ijerph-19-15356-f001:**
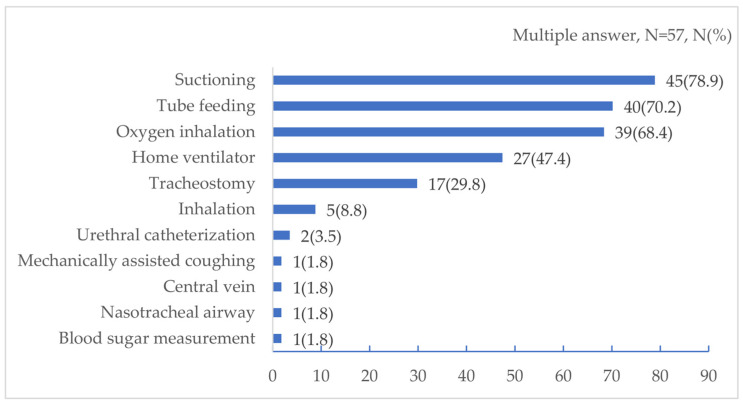
Medical devices and procedures required by the pediatric patients in this study.

**Table 1 ijerph-19-15356-t001:** Basic attributes, disaster preparedness, and intention to evacuate to a public shelter, by intention to evacuate (N = 57).

Item		Category	Total	Invention to Evacuate to Public Shelter		*p*-Value
				Yes	No	
			N (%)	N (%)	N (%)	
Patients with home medical care background						
Age (median [rang])	7.0 (0–46)					
Number of cohabitants (Mean [Standard deviation])	3.9 (0.9)					
Age group		Younger than 15 years	48 (84.2)	21 (43.8)	27 (56.3)	0.969 ^a^
		15 years or older	9 (15.8)	4 (44.4)	5 (55.6)	
Sex		Men	29 (50.9)	12 (41.4)	17 (58.6)	0.701 ^a^
		Women	28 (49.1)	13 (46.4)	15 (53.6)	
Number of cohabitants		Less than 4	16 (28.1)	11 (68.8)	5 (31.3)	0.018 ^a^
		4 or more	41 (71.9)	14 (34.1)	27 (65.9)	
Necessary medical devices and procedures						
Home ventilator		Yes	27 (47.4)	8 (29.6)	19 (70.4)	0.040 ^a^
		No	30 (52.6)	17 (56.7)	13 (43.3)	
Tracheostomy		Yes	17 (29.8)	8 (47.1)	9 (52.9)	0.751 ^a^
		No	40 (70.2)	17 (42.5)	23 (57.5)	
Oxygen inhalation		Yes	39 (68.4)	16 (41.0)	23 (59.0)	0.526 ^a^
		No	18 (31.6)	9 (50.0)	9 (50.0)	
Suctioning		Yes	45 (78.9)	18 (40.0)	27 (60.0)	0.255 ^a^
		No	12 (21.1)	7 (58.3)	5 (41.7)	
Tube feeding		Yes	40 (70.2)	20 (50.0)	20 (50.0)	0.152 ^a^
		No	17 (29.8)	5 (29.4)	12 (70.6)	
Urethral catheterization		Yes	2 (3.5)	2 (100)	0 (0.0)	0.188 ^b^
		No	55 (96.5)	23 (41.8)	32 (58.2)	
Blood sugar measurement		Yes	1 (1.8)	0 (0.0)	1 (100)	0.439 ^b^
		No	56 (98.2)	24 (42.9)	32 (57.1)	
Nasotracheal airway		Yes	1 (1.8)	0 (0.0)	1 (100)	0.000 ^b^
		No	56 (98.2)	25 (44.6)	31 (55.4)	
Central parenteral nutrition		Yes	1 (1.8)	0 (0.0)	1 (100)	0.000 ^b^
		No	56 (98.2)	25 (44.6)	31 (55.4)	
Inhalation		Yes	5 (8.8)	4 (80.0)	1 (20.0)	0.157 ^b^
		No	52 (91.2)	21 (40.4)	31 (59.6)	
Mechanically assisted coughing		Yes	1 (1.8)	0 (0.0)	1 (100)	1.00 ^b^
		No	56 (98.2)	25 (44.6)	31 (55.4)	
Background of Family Caregivers						
Age		20s	3 (5.3)	1 (33.3)	2 (66.7)	1.00 ^b^
		30s and older	54 (94.7)	24 (44.4)	30 (52.6)	
Sex		Men	1 (1.8)	0 (0.0)	1 (100)	1.00 ^b^
		Women	56 (98.2)	25 (44.6)	31 (55.4)	
Employment		Yes	20 (35.1)	5 (25.0)	15 (75.0)	0.035 ^a^
		No	37 (64.9)	20 (54.1)	17 (45.9)	
Regular employment		Yes	9 (15.8)	2 (22.2)	7 (77.8)	
Non-regular employment		Yes	11 (19.3)	3 (27.3)	8 (72.7)	
Previous experience with evacuation from a disaster		Yes	5 (8.8)	4 (80.0)	1 (20.0)	0.157 ^b^
		No	52 (91.2)	21 (40.4)	31 (59.6)	
Preparation for a disaster						
Prepared for a disaster		Yes	14 (24.6)	4 (28.6)	10 (71.4)	0.184 ^a^
		No	43 (75.4)	21 (48.8)	22 (51.2)	
Prepared children’s clothes		Yes	11 (19.3)	4 (36.4)	7 (63.6)	0.739 ^b^
		No	46 (80.7)	21 (45.7)	25 (54.3)	
Prepared food for the Patient with Home Medical Care (PHMC)		Yes	18 (31.6)	7 (38.9)	11 (61.1)	0.607 ^a^
		No	39 (68.4)	18 (46.2)	21 (53.8)	
Equipped with medical hygiene materials for the PHMC		Yes	16 (28.1)	6 (37.5)	10 (62.5)	0.546 ^a^
		No	41 (71.9)	19 (46.3)	22 (53.7)	
Prepared children’s medicine		Yes	7 (12.3)	4 (57.1)	3 (42.9)	0.687 ^b^
		No	50 (87.7)	21 (42.0)	29 (58.0)	
Secured a power source for continued use of medical devices during power outages		Yes	35 (61.4)	12 (34.3)	23 (65.7)	0.066 ^a^
		No	22 (38.6)	13 (59.1)	9 (40.9)	
Destination, preparation and intention of evacuation						
Secured shelter other than public shelter		Yes	20 (35.1)	5 (25.0)	15 (75.0)	0.035 ^a^
		No	37 (64.9)	20 (54.1)	17 (45.9)	
Will evacuate to welfare shelter		Yes	47 (82.5)	23 (48.9)	24 (51.1)	0.160 ^b^
		No	10 (17.5)	2 (20.0)	8 (80.0)	
Have simulated and prepared for life during evacuation		Yes	9 (15.8)	6 (66.7)	3 (33.3)	0.161 ^b^
		No	48 (84.2)	19 (39.6)	29 (60.4)	
Have asked others to help our evacuation		Yes	19 (33.3)	6 (31.6)	13 (68.4)	0.186 ^a^
		No	38 (66.7)	19 (50.0)	19 (50.0)	
Registered to the local government emergency evacuation assistance registry		Yes	25 (43.9)	11 (44.0)	14 (56.0)	0.985 ^a^
		No	32 (56.1)	14 (43.8)	18 (56.3)	
Knowledge of public shelter and decentralized evacuation						
I know where the public shelter is		Yes	51 (89.5)	20 (39.2)	31 (60.8)	0.077 ^b^
		No	6 (10.5)	5 (83.3)	1 (16.7)	
I know where the welfare shelter is		Yes	11 (19.3)	2 (18.2)	9 (81.8)	0.090 ^b^
		No	46 (80.7)	23 (50.0)	23 (50.0)	
I know where the welfare shelter is		Yes	2 (3.5)	0 (0.0)	2 (100)	0.499 ^b^
		No	55 (96.5)	25 (45.5)	30 (54.5)	
I know that the government is calling for decentralized evacuation to prevent COVID-19 transmission		Yes	21 (36.8)	9 (42.9)	12 (57.1)	0.907 ^a^
		No	36 (63.2)	16 (44.4)	20 (55.6)	
Disaster risk in the dwelling						
I know the risk of river flooding in the area of my dwelling		Yes	46 (80.7)	19 (41.3)	27 (58.7)	0.508 ^b^
		No	11 (19.3)	6 (54.5)	5 (45.5)	
I know the risk of landslides in the area of my dwelling		Yes	42 (73.7)	18 (42.9)	24 (57.1)	0.799 ^a^
		No	15 (26.3)	7 (46.7)	8 (53.3)	
I know the risk of inundation by tsunami in the area of my dwelling		Yes	42 (73.7)	17 (40.5)	25 (59.5)	0.389 ^a^
		No	15 (26.3)	8 (53.3)	7 (46.7)	

^a^ χ^2^ test, ^b^ Fisher ‘s direct probability test.

**Table 2 ijerph-19-15356-t002:** Factors associated with evacuation intention: age, parents’ evacuation experience, use of home mechanical ventilation, and securing evacuation sites other than a public shelter.

Item	Category	OR	95% CI		*p*-Value
			Lower Limit	Upper Limit	
Age	Less than 15 years/15 years or older	1.27	0.26	6.29	0.774
Parents’ Evacuation Experience	Yes/No	11.51	0.77	171.78	0.077
Use of a home ventilator	No/Yes	3.99	1.13	14.03	0.031
Ensure evacuation destinations other than designated evacuation centers	No/Yes	7.29	1.62	32.88	0.010

Binomial logistic regression analysis; Inputted variables: Number of cohabitants, employment of parents; Abbreviations: CI: confidence interval, OR: odds ratio.

## Data Availability

The data analyzed during this study are included in this published article. Further inquiries can be directed to the corresponding authors.

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
