# Peer review of "Disaster Evacuation for Home-Based Patients with Special Healthcare Needs: A Cross-Sectional Study"

_ijerph, 2022, doi:10.3390/ijerph192215356_

Round 1

Reviewer 1 Report

The introduction should point out whether there is any similar previous studies and findings in these studies and what the present study has advanced the knowledge building on these previous studies. It should also position the study within the current discussion of health disaster and emergency risk management in the field.

Specifically, Lines 50-53 seem unrelated, if not contradictory, to the theme of the paper regarding intention to evacuate. As to Lines 78-80 and 87-90, the study's findings seem to suggest that COVID-19 did not affect the intention to evacuate, which should be pointed out in both the introduction and conclusion.

Lines 102-104 seem to suggest the problems of public shelter, which might be contradictory to the theme of investigating factors associated with intention to evacuate to a public shelter so as to encourage such evacuation.

In the Data collection section, it was said that "interview survey was conducted by one researcher and one outpatient nurse" (Lines 122-123), which seems inconsistent with "self-report questionnaire" (Line 129). As to sampling, the use of non-probabilistic sampling needs to be justified, including why sample should be selected from the Pediatrics Department of Fukuoka University Hospital (e.g., is it the biggest, receiving most of relevant patients, the best equipped?). The background of this hospital needs some description. 

In the Analytical methods section, the characteristics of the participants and family caregivers and the distribution of medical devices surveyed in this study should better be compared with those in the region or the country to allow assessing the value of this study's findings (Lines 177-180).

In the Results session, it is unclear whether the remaining 32 people out of the 89 refused to give responses or could not be reached. It may also be useful to compare the characteristics of these 32 people with the 57 people from whom responses were obtained. (Lines 209-211)

In the discussion section, it points out that there were previous studies about power-dependence. It will be helpful if power-dependence is incorporated and investigated in the questionnaire. (Lines 269-270)

In the same section, the effect of COVID-19 on intention to evacuate is discussed (Lines 298-311). However, this speculated effect is not supported by the study's findings. In the discussion of limitations, it was actually admitted that the impact of the pandemic was not known (Lines 319-320). This begs the question why the effect of COVID-19 was not investigated in this study, although there is one question "I know that the government is calling for decentralized evacuation to prevent COVID-19 transmission", which was found to have no significant association with intention to evacuate. Moreover, the statement "they are willing to evacuate because they have secured other evacuation sites" (Line 308) might be wrongly written.  

Lastly, Table 1 needs some spelling checking. For example, "decentralized evolution" on page 8 seems to be "decentralized evacuation".

Reviewer 2 Report

Minor amendments:

-          The title of the paper is too long and should be shortened.

-          Reference 34 Tanaka in the text should change to Soichiro as can be seen in the references and the reference should be completed. The title of the paper is not included at the moment.  

-          In discussion section, a sentence or two can be addressed regarding the reliability of the findings.

-          The title Conclusion can be rewritten in “conclusion and recommendations” as there are few suggestions from the author in this section.

Author Response

添付ファイルをご覧ください。

Round 2

Reviewer 1 Report

I highly appreciate the efforts the authors have put to address the issues and revise the manuscript in a sufficient way and I support its publication in IJERPH.